# Delivery of Immunostimulatory Cargos in Nanocarriers Enhances Anti-Tumoral Nanovaccine Efficacy

**DOI:** 10.3390/ijms241512174

**Published:** 2023-07-29

**Authors:** Jenny Schunke, Volker Mailänder, Katharina Landfester, Michael Fichter

**Affiliations:** 1Department of Dermatology, University Medical Center Mainz, Langenbeckstr. 1, 55131 Mainz, Germany; 2Max Planck Insitute for Polymer Research, Ackermannweg 10, 55128 Mainz, Germany

**Keywords:** nanovaccines, adjuvants, tumor-specific antigens, DC targeting, co-delivery

## Abstract

Finding a long-term cure for tumor patients still represents a major challenge. Immunotherapies offer promising therapy options, since they are designed to specifically prime the immune system against the tumor and modulate the immunosuppressive tumor microenvironment. Using nucleic-acid-based vaccines or cellular vaccines often does not achieve sufficient activation of the immune system in clinical trials. Additionally, the rapid degradation of drugs and their non-specific uptake into tissues and cells as well as their severe side effects pose a challenge. The encapsulation of immunomodulatory molecules into nanocarriers provides the opportunity of protected cargo transport and targeted uptake by antigen-presenting cells. In addition, different immunomodulatory cargos can be co-delivered, which enables versatile stimulation of the immune system, enhances anti-tumor immune responses and improves the toxicity profile of conventional chemotherapeutic agents.

## 1. Different Factors Establishing the Immunosuppressive Tumor Microenvironment

The effective treatment of cancer still holds many challenges due to the heterogeneity of tumors in patients. Moreover, different mechanisms of the immune system are affected by tumor cell alterations. For example, the downregulation or loss of HLA class I/MHC class I expression or defects in the antigen-processing machinery in antigen-presenting cells (APCs) [1,2] is affected, which in turn leads to impaired T cell activation against tumors. The expression of immune checkpoint ligands, such as programmed death-ligand 1 (PD-L1), by tumor cells and the secretion of inhibitory cytokines, e.g., TGF-β, can also inhibit the function of APCs [3,4]. The binding of PD-L1 to programmed cell death protein 1 (PD-1) not only inhibits dendritic cells (DCs) but also T cells directly and thereby suppresses their activation. Additionally, signaling through other expressed immune checkpoints, such as cytotoxic T lymphocyte-associated protein 4 (CTLA-4), lymphocyte-activation gene 3 (LAG3) or T cell immunoglobulin and mucin domain 3 (TIM-3) suppresses the function of immune cells in the tumor microenvironment (TME) [5,6]. To circumvent this immunosuppression, immune checkpoint-blocking antibodies have been successfully used in clinics. In particular, patients with advanced melanoma have benefited from therapy with monoclonal anti-PD-1 antibodies (nivolumab) or a combination of PD-1 and CTLA-4 (ipilimumab) blockade [7].

Nevertheless, the TME is often composed of immunosuppressive cells, such as regulatory T cells (T_regs_), myeloid-derived suppressor cells (MDSCs) or inhibitory (M2-type) macrophages (Figure 1) [8,9,10]. FoxP3^+^ T_regs_ not only inhibit the differentiation of naïve T cells to effector cells but also inhibit the function of CD4^+^ and CD8^+^ T cells as well as of NK cells, B cells and DCs [11,12]. MDSCs are able to suppress T cell activity with the production of ROS [10] and the expression of arginase and iNOS [13,14]. It was further shown that MDSCs promote the differentiation of FoxP3^+^ T_regs_ in vivo [15,16]. Immunosuppression by M2-type macrophages is based on the release of anti-inflammatory molecules subsequently promoting tumor growth [17].

To overcome immune evasion and to induce tumor-specific T cell responses, immunotherapeutic vaccines are designed based on tumor antigens [18]. Those tumor-associated antigens (TAA) and tumor-specific antigens (TSA) offer different advantages in their use as vaccine components in terms of prevalence, T cell specificity and generation of immune tolerance or autoimmunity [19,20,21]. This review provides an overview of various antigen-based vaccination formulations and the extent to which nanomedicine can overcome existing challenges. In particular, the focus is on the advantages of nanocarriers, such as the protected transport of cargos and the resulting extended circulation time, as well as the possibility of an all-in-one delivery of antigens, adjuvants and drugs. In addition, defined quantities of transported cargos can be controlled, and non-specific diffusion of small molecules can be prevented. Furthermore, the improvement of toxicity profiles of chemotherapeutic drugs by their encapsulation into nanocarriers is discussed.

## 2. Nucleic Acid-Based Vaccines: DNA and RNA Encoding for Tumor Antigens

Nucleic acid-based vaccines consist of DNA or RNA encoding for TAA as well as for TSA [22,23]. Research efforts have focused on DNA vaccines due to their ease of production and stability during storage. They can be designed by incorporation of desired sequences into a plasmid backbone [24]. Furthermore, DNA vaccines offer a way to mimic viral infections [25] by DNA binding to Toll-like receptors and thereby induce proinflammatory immune responses and can be flexibly adapted by genetic modifications [24]. However, they have shown unsatisfactory results in clinical trials due to low uptake in antigen-presenting cells and the resulting inefficient expression of antigens [26]. Nevertheless, tumor-specific T cell and IgG responses could be generated with DNA fusion vaccines in pre-clinical studies [27] using electroporation (EP) [28,29]. For instance, an improved response to an HIV-1 DNA vaccine was induced using EP as the application method [30].

Since mRNA is translated in the cytoplasm of a target cell, mRNA vaccines do not need to enter the nucleus, in contrast to DNA vaccines [22]. On the other hand, RNA has a lower stability and is rapidly degraded in biological fluids [31]. Therefore, mRNA vaccine design focuses on the increase of RNA half-life by optimizing the 5′- and 3′-UTR elements via genetic modifications or the encapsulation in delivery vehicles [32,33]. Non-formulated mRNA is mainly taken up by immature DCs, thus, so-called “naked” mRNA is administered intradermally or intranodally [34,35]. Even though naked RNA has induced antigen-specific T cell responses in pre-clinical studies [36], mRNA stability is challenging. One way to circumvent mRNA instability is to load dendritic cells (DCs) ex vivo, which in turn is time-consuming and expensive [37]. Electroporation of DCs with mRNA encoding for CD70, CD40-L, and constitutively active Toll-like receptor 4 (caTLR4) has induced effective DC maturation and subsequent T cell stimulation [38]. This so-called TriMix-RNA was combined with additional mRNAs, each encoding for one of four melanoma-associated antigens (MAGE-A3, MAGE-C2, tyrosinase, or gp100), and further introduced into DCs by electroporation. In clinical trials for the treatment of stage III/IV melanoma patients, this vaccine has been shown to be safe and immunogenic and can further be improved with regard to long-term immunity by being combined with an immune checkpoint blockade [39,40]. Treatment with the TriMix/mRNA vaccine in combination with ipilimumab resulted in an overall survival of 28% and a progression-free survival of 18% after more than 5 years.

## 3. Tumor Cell-Based Vaccines

Early vaccination approaches focused on the application of whole cells or cell lysates for antigen delivery [41]. Designing vaccines based on autologous and allogeneic tumor cells offers the advantage that tumor antigens do not have to be identified in advance by DNA/RNA sequencing techniques. In addition, these vaccines contain a wide range of tumor antigens, which can thus generate a broad immune response. To improve the immunogenicity of whole cell vaccines, tumor cells can be genetically modified to express cytokines and chemokines. The GVAX cancer vaccine, first developed in 1993 by Glenn Drandoff, consisting of two replication-deficient prostate carcinoma cell lines, which were genetically modified to secret GM-CSF, were tested in clinical trials [42,43]. This therapy for advanced prostate cancer was well tolerated and prolonged overall survival dose-dependently. Improved effects were achieved treating patients with advanced melanoma using a polyvalent melanoma vaccine consisting of three irradiated human melanoma cell lines [44,45]. Intradermal injection of this melanoma vaccine significantly increased the overall survival of stage IIIA and IV melanoma patients by three- or fourfold, respectively. Other approaches, such as the cancer vaccine Melacine, combine allogeneic melanoma cell lysates with adjuvants [46,47]. This vaccination strategy induced modest anti-tumoral effects in clinical studies and induced the strongest anti-tumor activity in patients expressing the HLA class I antigens A2 or C3 by most efficient induction of CD8^+^ T cell responses.

## 4. Dendritic Cell-Based Vaccines

Since dendritic cells (DCs) can prime naïve T cells in an antigen-specific manner, various DC-based vaccines have been explored [48,49]. Following antigen uptake, DCs maturate, migrate into the lymph nodes and present antigenic peptides bound to MHC class I and II molecules to T cells [50,51]. T cell priming and proliferation is based on three DC-based signals: (i) T cell receptor (TCR) binding to the antigen/MHC-complex, (ii) binding of the costimulatory receptors CD80 and CD86 expressed by DCs, and (iii) cytokine signaling [52,53]. In various studies evaluating the efficacy of DC-mediated vaccines, monocyte-derived DCs cultured with GM-CSF and IL-4 were used. Prior to immunization, they were loaded with tumor antigens ex vivo, such as MHC class I-restricted peptides, synthetic long peptides or full-length proteins [54,55,56]. Early clinical trials for the treatment of melanoma patients describe the pulsing of in-vitro-generated DCs with either a cocktail of melanoma-associated peptides (tyrosinase, Melan-A/MART-1, gp100) or peptides derived from MAGE-1 and MAGE-3 [57]. Those peptide-loaded DCs were repeatedly injected intralymphatically depending on the patient’s response to the vaccination. Another group of patients was injected with tumor lysate-pulsed DCs. In this study, the induction of DC vaccine-mediated antigen-specific T cell activity against melanoma and metastases in different organs could be observed. The suitability of antigen-pulsed DCs was further confirmed in a B cell lymphoma vaccination trial [58] as well as for the treatment of acute myeloid leukemia [59] and myeloma [60]. Furthermore, autologous peptide-loaded DCs were tested for their potential to induce anti-melanoma immune responses [41,61]. DCs were loaded with MHC class I- and II-restricted peptides and injected subcutaneously. However, there was no increased response rate or overall survival compared to standard chemotherapy with the cytostatic agent dacarbazine.

## 5. Adjuvants Play a Key Role in Enhancing Immune Responses to Vaccines

Adjuvants are immunomodulatory molecules enhancing antigen-specific immune responses. In this way, they improve the antigen-directed response to vaccines, strengthen the durability of the immune response to vaccine stimuli or trigger a more extensive immune response [62,63]. In the 1920s, aluminum salts were first approved for application as vaccine adjuvants in humans [64]. Aluminum hydroxide and aluminum phosphate are still important adjuvants present in various licensed vaccines. Their effect is based on the stimulation of dendritic cells (DCs), the activation of the complement system and the induction of chemokine production [65,66,67]. However, they cannot elicit antigen-specific CD8^+^ T cell and T_h_1 responses and generally enhance T_h_2-mediated antibody-based immune responses, which are not sufficient for robust tumor killing [68]. Since then, the development of novel adjuvants is steadily progressing. Nowadays, a broad range of clinically tested and used adjuvants for vaccination approaches is available [69]. Besides aluminum salts, these include adjuvant-containing emulsions, virosomes, dsRNA analogs, lipid A analogs or imidazoquinolines [69].

TLRs represent important adjuvant targets for detecting pathogen-associated molecular patterns and are mainly expressed by antigen-presenting cells. TLR4 is localized in the plasma membrane, while TLR7/8 and 9 are located in endosomal membranes [70]. The immunomodulatory potential of TLR agonists has been widely used in the testing and development of adjuvants for vaccination.

It has been shown that the TLR3 and MDA5 agonist Poly(I:C) induces the production of type I interferons and other pro-inflammatory cytokines, subsequently enhancing T cell activity and proliferation [71]. Furthermore, CpG oligodeoxynucleotides, which interact with TLR9, primarily stimulate B cells, T cells as well as natural killer (NK) cells and macrophages [72]. In addition to those polymer-like adjuvants, small molecules, such as imidazoquinolines, bind to TLR7 and TLR8, which play an important role in the induction of anti-viral immune responses by naturally recognizing single-stranded RNA [3,73,74]. The imidazoquinoline resiquimod (R848) was shown to activate the MyD88 signaling pathway by binding to TLR7 or TLR8 and subsequently inducing the secretion of pro-inflammatory cytokines by NF-ϰB-mediated transcription [75]. Due to this property, R848 became a promising adjuvant not only for vaccination against pathogens but also for use in cancer vaccines. Clinical studies have shown an improvement of pancreatic tumor control by combining radiotherapy and R848 application [76]. Furthermore, this combination treatment elicited an anti-tumor immune response in pre-clinical studies against melanoma [77]. In addition, the combination of the TLR3 agonist Poly(I:C) and the TLR7/8 agonist R848 enhanced the polarization of macrophages to inflammatory (M1-like) effectors in vitro and induced T cell infiltration followed by tumor regression in murine lung cancer and fibrosarcoma models [76]. Additionally, single-stranded RNA with uridine- and guanosine-rich sequences can also act as TLR7/8 agonists and thereby promote Th1 responses and the secretion of IFN-α and IL-12 as an adjuvant.

Multiple studies have demonstrated a correlation of induced high levels of type I interferons upon anti-cancer immunotherapy with a better outcome [78,79]. Therefore, agents triggering the activation of the stimulator of interferons genes (STING) came into the focus [78,79]. STING, a transmembrane protein located in the endoplasmic reticulum, plays an important role in the sensing of cytosolic DNA, which triggers the cGAS/STING pathway [53]. This leads to the downstream production of type I interferons affecting T cells, NK cells [80,81], APCs [82,83] and tumor cells themselves. It is known that those interferons inhibit the proliferation of tumor cells and induce the expression of MHC class I, while the expression of VEGF is reduced [84,85,86]. First generation STING agonists, such as DMXAA, significantly reduced tumor growth but failed to overcome immunosuppressive TME and did not induce long-term immunity in mouse models [87,88]. Even though DMXAA was successfully applied in pre-clinical studies and was well tolerated in clinical trials, it failed to prolong the overall survival of non-small-cell lung cancer patients compared to placebo treatments [89,90]. Those contrary results in mouse models and clinical trials can be explained by polymorphisms in human STING, which prevent effective binding of DMXAA in many patients rendering therapy with this STING agonist ineffective [91]. This finding led to the development of synthetic cyclic dinucleotides, such as ADU-100. Intratumoral injection of ADU-100 was shown to induce antigen-specific activation of CD8^+^ T cells and to improve cancer therapy with antibodies specific for the immune checkpoints PD-1 and CTLA-4 [92,93,94,95]. Next-generation non-cyclic dinucleotides, such as ALG-031048, with higher stability were further developed. Intratumoral application of ALG-031048 increased the regression rate of CT26 colon tumors from 44% following treatment with ADU-100 to 90%. It additionally promoted an effective long-term immune memory in mice [96]. Nevertheless, the use of these STING agonists was limited by their low stability and their systemic administration was not feasible. Therefore, a new class of STING agonists, amidobenzimidazoles (ABZI), with a higher stability and increased potency were developed. In pre-clinical trials, ABZI-based compound 3 (diABZI) induced a 400-fold stronger IFN-β production compared to the natural STING agonist cGAMP. Furthermore, the systemic treatment of murine CT26 tumors led to an effective anti-tumoral immune response based on CD8^+^ T cells [97,98].

## 6. Nanomedicine Enables the Combined Delivery of Immunostimulatory Cargos and Reduces Side Effects Elicited by Chemotherapeutic Drugs

Using nanoparticles (NPs) as delivery vehicles for antigens, adjuvants or drugs ensures their protected transport, prolonged bioavailability and controlled release [99]. In addition, different nanocarrier groups can be selected for specific applications, as they differ not only in composition, but also in loading capacity, size, shape and surface charge (Figure 2) [100,101,102,103]. When used as vaccine formulations in cancer immunotherapy, the uptake of NPs by dendritic cells (DCs) is particularly important to ensure tumor antigen-specific activation of the immune system. DC uptake cannot exclusively be influenced by NP properties but can also be increased by specific modification of the particle surfaces. These modifications include the conjugation of antibodies [104] or other targeting moieties [105].

NPs composed of inorganic materials are of interest for application as tumor vaccines due to their stability in biological fluids and their controllable synthesis (Figure 3) [106]. In addition, depending on the material from which they are synthesized, they inherit various advantages and disadvantages.

Gold nanoparticles, for example, were shown to stimulate the immune system by inducing different cytokine pathways. This immune-system-activating potential is dependent on the size and shape of the NPs [107].

Silica-based NPs are promising inorganic formulations due to their non-toxic profile and biodegradability [108,109]. In vitro studies have demonstrated the successful encapsulation of dexamethasone into core-shell silica nanocapsules for the treatment of liver diseases [110]. Encapsulation of drugs can also enhance their solubility, stability and reduce side effects. This was shown for the encapsulation of four different chemotherapeutic drugs (cisplatin, carboplatin, oxaliplatin, and oxalipalladium) into silica nanocapsules [111]. Fan et al. additionally demonstrated the efficient covalent conjugation of the anti-cancer drug doxorubicin and folic acids to the NP surface [112]. Surface modifications enhanced the NP uptake by folate-receptor-expressing cancer cells and reduced cytotoxicity due to lower drug release levels in folate-receptor-negative cells. However, silanol groups of silica NP surfaces can interact with phospholipids of red blood cells and thereby induce hemolysis [113]. Those disadvantages can reduce their applicability in vivo. Thus, other inorganic nanoparticles, such as carbon nanospheres, solid carbon nanoparticles or carbon nanotubes consisting of graphite layers came into focus [114,115]. Their core-shell morphology provides a large loading space and can be used for the encapsulation of drugs or immune checkpoint inhibitors (ICIs) [114]. Additionally, the biocompatibility of carbon NPs enables oral vaccine administration [115]. In addition, encapsulation protects cargos against enzymatic degradation in the gastrointestinal tract, which even allows oral administration of unstable molecules [116].

Liposomes made of biodegradable phospholipids are uni-, bi- or oligolamellar vesicles which offer another option for effective encapsulation of immunomodulatory compounds [117]. They were first introduced in 1965 [118] and were used for vaccine development in 1974 [119]. Since various parameters, such as size, charge, surface modification and loading are variably adjustable, they represent versatile delivery vehicles for adjuvants and antigens (Figure 4) [120]. In particular, the surface charge can modulate uptake in tissues and cells such as APCs. Cationic liposomes, for example, interact with DC surfaces due to their positive zeta potential, which enhances their uptake, and further induces DC maturation [121,122]. These properties also allow an application by various routes, such as oral, topical or mucosal administration. Cargos can be encapsulated into the hydrophilic core of liposomes, embedded into the lipid bilayer or attached to the surface via modification of acyl chains or complexation [120]. An example of DC-stimulatory liposomes are RNA-lipoplexes (RNA-LPX) synthesized by complexing antigen-encoding RNA with liposomes [123]. Since single-stranded RNA naturally binds to TLR7 and TLR8, RNA-LPX induces DC maturation and thereby leads to the production of pro-inflammatory cytokines and T cell activation. This has also been proven in pre-clinical studies in which the vaccination of CT26 colon-tumor-bearing mice with RNA-LPX induced strong anti-tumoral cellular and humoral immune responses. Intravenously injected RNA-LPX was further described as a well-tolerated treatment for melanoma patients and offers the opportunity of personalized cancer treatment [124]. Vaccine-mediated and dose-dependent production of IFN-α and antigen-specific T cell responses were observed. Protected delivery of mRNA in lipid nanoparticles (LNP) was additionally demonstrated by preventative immunization with COVID-19 mRNA vaccines [125]. This approach also allows the complexation of mRNA encoding tumor antigens or therapeutic antibodies [126]. LNP consists of ionizable cationic lipids, phospholipids, lipids attached to polyethylene glycol (PEG) and cholesterol. Ionizable lipids are needed for mRNA complexation, whereas cholesterol and other helper lipids improve LNP stability [126,127]. Surface PEGylation further enhances the LNP circulation time [127]. Cationic 1,2-dioleoyl-3-trimethylammonium-propane (DOTAP)-based LNP were shown to interact with serum proteins and thus aggregated, resulting in a short half-life. Furthermore, their hemolytic activity induced severe side effects. Therefore, ionizable LNP with an improved toxicity profile have been developed for pH-sensitive mRNA delivery. Different mRNA-loaded LNP are being evaluated for their efficacy in clinical trials for the treatment of tumors, such as lymphoma or melanoma. The packaged mRNA encodes target proteins, such as human IL-12, OX40L, or for different neoantigens. Moreover, additive treatment effects are tested by combining the mRNA LNP with monoclonal antibodies blocking immune checkpoints [126].

Further promising carrier systems for use as anti-cancer vaccines are micelles. They enable the efficient co-encapsulation of antigens and adjuvants and, thus, enhance DC-mediated antigen-specific T cell activation [128]. This was demonstrated via the encapsulation of ovalbumin (OVA) and the TLR7 agonist CL264 into micelles based on amphiphilic diblock co-polymers [129]. Vaccination with these micelles enhanced antigen cross-presentation of DCs to CD8^+^ T cells, resulting in E.G7-OVA tumor growth prevention in vivo. Similar results were obtained by Zeng et al. with the melanoma antigen TRP2 and TLR9 agonist CpG ODN-loaded self-assembled micelles based on two amphiphilic diblock co-polymers [130]. Their application in vivo led to strong anti-tumoral immune responses mediated by cytotoxic T lymphocytes in a lung metastatic melanoma model. Moreover, epirubicin-conjugated micelles have been successfully tested for application in humans and showed decreased side effects compared to conventional epirubicin administration in the treatment of solid tumors [131]. The treatment of patients with Paclitaxel (PTX)-loaded polymeric micelles (NK105) was compared to soluble PTX in a phase III clinical trial and their efficacy regarding breast cancer therapy was evaluated [132,133]. The efficacy of the NK105 formulation was comparable with regard to the overall survival and was less toxic, as evidenced by the occurrence of peripheral sensory neuropathy. Genexol^®^-PM consisting of PTX-loaded micelles was approved in South Korea in 2007 for the treatment of breast cancer, ovarian cancer as well as for non-small-cell lung cancer (NSCLC) [134,135].

The synthesis of polymeric NPs for vaccination purposes has been extensively researched [136]. Different vaccines composed of poly(D,L-lactic-co-glycolic acid) (PLGA)-based NPs were developed and their potential to transport encapsulated bioactive cargos specifically to DCs was verified [137]. The uptake of PLGA NPs was detected by both murine and human cells [138,139,140], with the uptake rate being highest for cationic NPs [141]. In vivo studies demonstrated that NP size below 500 nm is beneficial for the uptake and subsequent activation of cytotoxic T lymphocytes. Small NPs are preferentially taken up by DCs and larger ones by macrophages, which explained these observations [137,142]. PLGA-based NPs can be loaded with antigens as well as adjuvants. This allows the co-delivery of multiple adjuvants, such as TLR agonists [143], as well as the reduction of the adjuvant amount needed for robust DC-mediated T cell priming [144]. Diwan et al. immunized BALB/c mice with CpG-loaded PLGA-NPs and showed that the amount of CpG required for T cell activation could be reduced by 10- to 100-fold by encapsulation into NPs. Further in vivo studies additionally demonstrated the induction of antigen-specific T cell responses by the encapsulation of antigens and the simultaneous enhancement of immune responses by encapsulated adjuvants [145,146]. The combined encapsulation of OVA and the TLR4 ligand monophosphoryl lipid A induced antigen-specific T cell activation as well as a strong production of the pro-inflammatory cytokine IFN-γ. IFN-γ plays an important role in anti-cancer immunity by triggering the expression of MHC class I and II molecules on DCs [147] resulting in enhanced antigen presentation. Furthermore, B16/F10 melanoma-bearing mice could be efficiently treated with TRP2/7-acyl lipid A-loaded PLGA NPs. Vaccination with those PLGA-NPs triggered the production of different pro-inflammatory cytokines, such as IL-6, IL-12, TNF-α and IFN-γ, as well as strong T cell-mediated reduction of tumor volume [148]. As an alternative to PLGA, other copolymers can be utilized for the synthesis of polymeric nanoparticles. Amphiphilic hybrid and fully synthetic copolymers, such as poly(ethylene glycol), polyoxazolines, synthetic glycopolymers or hydrophilic poly(amino acids) are used as hydrophilic blocks [149]. Polycarbonate, polystyrene or aliphatic polyesters (e.g., polycaprolactone and poly(lactic acid)) are used as hydrophobic blocks [149].

The synthesis of protein-based nanocarriers for medical use is of particular interest due to their good biocompatibility as well as their biodegradability. Protein nanocapsules (NCs) were efficiently synthesized from bovine serum albumin (BSA) and ovalbumin (OVA) [150] and showed a strong uptake by DCs. Cationic BSA can further be used to form complexes with siRNA. Nanoparticles based on cationic BSA and Bcl2-specific siRNA were used for the treatment of mice with lung metastasis [151]. This vaccine formulation exhibited low toxicity and efficiently inhibited tumor growth. Encapsulation of antigens and adjuvants in E2 protein nanoparticles showed efficient activation of DCs and T cells in vitro and in vivo [152]. E2 proteins, derived from a subunit of the pyruvate dehydrogenase complex from bacteria, self-assemble into nanoparticles whose interfaces can be modified for further site-directed functionalization [153,154]. Loading of E2 nanoparticles with SIINFEKL peptide and CpG ODN triggered antigen cross-presentation by DCs and subsequent T cell activation in vitro [155]. Combined encapsulation of the TAA gp100 with CpG ODN as an adjuvant induced strong CD8^+^ T cell proliferation in vivo as well as enhanced IFN-γ production. The efficiency of antigen and adjuvant co-delivery in E2 NPs was demonstrated by treating B16/F10 melanoma-bearing mice, with regard to prolonged overall survival [156]. Moreover, the importance of cargo co-delivery for efficient DC-mediated cancer therapy was further shown by Hüppe et al. [157]. This study demonstrated the feasibility of the encapsulation of three adjuvants with different solubility in nanocapsules (NCs) composed of human serum albumin (HSA). Dendritic cells showed the strongest activity in terms of the expression of CD80 and CD86 after the uptake of NCs containing all three adjuvants: Poly(I:C), R848 and muramyl dipeptide. This observation additionally demonstrated the importance of simultaneous cargo encapsulation and delivery, which causes extensive DC activation and consequently induces an anti-tumoral immune response. Moreover, proteins derived from milk or corn can also serve as nanoparticle shell material. Zein, a storage protein present in corn, can be used as a biocompatible source for the synthesis of nanocarriers. For breast cancer treatment, Zein nanoparticles were loaded with the chemotherapeutic drug 5-fluorouracil [158]. Other studies describe the encapsulation of cisplatin into casein-based nanoparticles [159], inducing stronger anti-tumoral immune responses against hepatic tumors in a mouse model compared to the conventional cisplatin treatment. They further penetrated the tumor tissue allowing cisplatin to exert its effect or to be taken up by cells directly within the tumor. To achieve targeted uptake of loaded nanoparticles into tumor cells, not only can the particle surfaces be modified, but specific shell materials can also be selected. Metabolically active tumors often express lactoferrin receptors at high levels. Golla et al. took advantage of this property by synthesized lactoferrin-based nanoparticles loaded with doxorubicin [160]. Oral administration of those drug-loaded lactoferrin-NPs significantly reduced the growth of HCC tumor nodules in mice compared to the treatment with soluble doxorubicin. Abraxane^®^ is an FDA-approved nanoparticle formulation consisting of albumin-bound paclitaxel for the treatment of breast cancer [161]. A phase III clinical trial demonstrated improved efficacy and reduced side effects in breast cancer patients treated with Abraxane^®^ compared to standard paclitaxel application [162].

## 7. Combining Nanovaccines with Immune Checkpoint Therapy Enhances Anti-Tumoral Immune Responses

To further enhance the efficacy of nanovaccines, different combination studies with immune checkpoint inhibitors (ICI) were performed. Liu et al. combined aerosolized nanoparticles (NPs) containing cyclic dinucleotides with anti-PD-L1 antibodies for the treatment of murine non-small-cell lung cancer [163]. This combination therapy not only induced robust CD8^+^ T cell activation through STING stimulation but also reduced T cell inhibition by the PD-L1 blockade. Furthermore, a reprogramming of anti-inflammatory macrophages to pro-inflammatory macrophages was induced, indicating an anti-tumorigenic phenotype. Another pre-clinical study demonstrated enhanced anti-tumoral immune responses by combining platinum-complex-loaded PC7A-NPs with an immune checkpoint blockade [164]. Nanoparticles released the encapsulated platinum complex pH-dependently in the tumor microenvironment and C7A monomers subsequently acted as the adjuvant. By combining this nanovaccine with ICI, CT26 colon tumor growth was strongly inhibited. Similar results were obtained by co-encapsulating the chemotherapeutic drugs paclitaxel and chloroquine with the antigen ovalbumin, the adjuvant CpG ODN as well as anti-PD-L1 antibodies into polymeric nanoparticles [165]. This combination therapy was efficiently tested in pre-clinical tumor models and induced a long-term immune memory against the encapsulated antigen. Further in vivo studies demonstrated an enhanced vaccination potency by combined encapsulation of TRP2-coding mRNA and PD-L1 siRNA into lipid-coated calcium phosphate NPs for the treatment of murine B16F10 melanoma [166]. The NP-induced knockdown of PD-L1 enhanced the antigen-specific antitumor immune response. Very recently, the treatment of patients with advanced squamous cell carcinoma could also be improved by combining a neoantigen-based vaccine with anti-PD-1 antibodies in a clinical trial [167].

## 8. Conclusions and Future Perspectives

Immunotherapies aim to activate the immune system in a tumor-specific manner and thereby overcome the immunosuppressive features of the tumor microenvironment. Various approaches for preventive and therapeutic vaccination have been tested pre-clinically and clinically. Nevertheless, many therapeutic approaches do not achieve complete or long-term tumor remission. The encapsulation of adjuvants, antigens, chemotherapeutic drugs and immune checkpoint inhibitors enhances the activation of dendritic cells. In particular, stimulation of dendritic cells can be achieved by the simultaneous delivery of encapsulated antigens and adjuvants, resulting in improved T cell activation. In the future, the combination of nanovaccines and an immune checkpoint blockade will provide extensive potential to address the immune system in different ways. In addition, nanocarrier-based vaccine formulations offer the opportunity to personalize cancer therapy by the encapsulation of pre-screened neoantigens. The encapsulation of patient-specific tumor peptides or mRNA coding for those peptides is a promising approach to efficiently treat cancer patients and to achieve prolonged overall survival. The functionalization of NP surfaces also offers an opportunity for more specific targeting of antigen-presenting cells such as DCs. Mannose functionalization or conjugation of receptor-specific antibodies onto NP surfaces and the coupling of nanobodies to nanoparticulate carrier systems can be used for this purpose. Since nanoparticles are versatile and modifiable, it will be of particular interest in the future to combine all the knowledge gained so far, so that antigens, adjuvants, ICI and cell targeting are combined in one NP-based vaccine subsequently influencing various mechanisms of the immune system. In addition, the establishment of various NP classes also enables needle-free administration (e.g., oral or intranasal administration), which will also bring advantages in the future, for example, in the vaccination of children or patients with needle phobia. The upscaling of different particle formulations is particularly challenging. NPs, such as micelles, polymer-based NPs and solid lipid NPs, are suitable for large-scale production due to their physico-chemical properties, ease of production and stability [168,169]. However, batch-to-batch variability, sterile production, and the provision and cleaning of suitable equipment are the main challenges faced by the industry in the future [170,171]. Additionally, controlling particle size and shape is not possible with every synthesis method used in laboratories for larger-scale approaches [171]. Nevertheless, methods, such as high-pressure homogenization (HPH), hot melt extrusion in combination with HPH, microemulsion techniques, nanoprecipitation and microchannels enable the synthesis of NPs on a large scale [169].

## Figures and Tables

**Figure 1 ijms-24-12174-f001:**
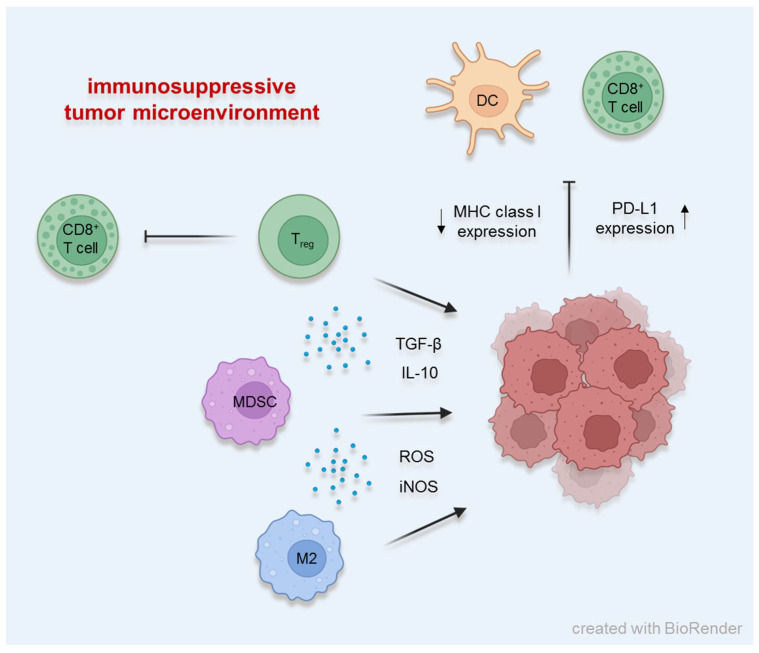
Cellular mechanisms maintaining the immunosuppressive tumor microenvironment.

**Figure 2 ijms-24-12174-f002:**
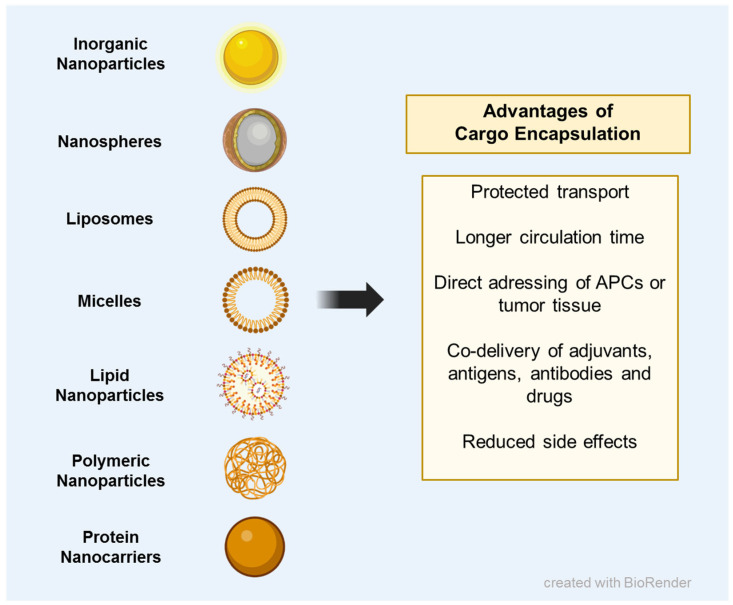
Encapsulation of biomedical cargos into nanocarriers increases anti-tumoral nanovaccine efficacy.

**Figure 3 ijms-24-12174-f003:**
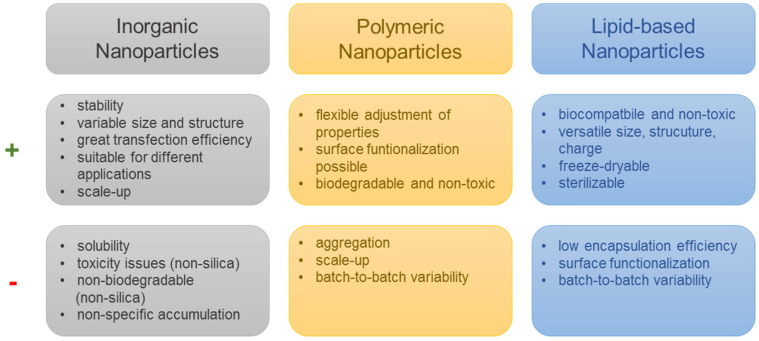
Advantages (+) and challenges (−) of different nanoparticle classes.

**Figure 4 ijms-24-12174-f004:**
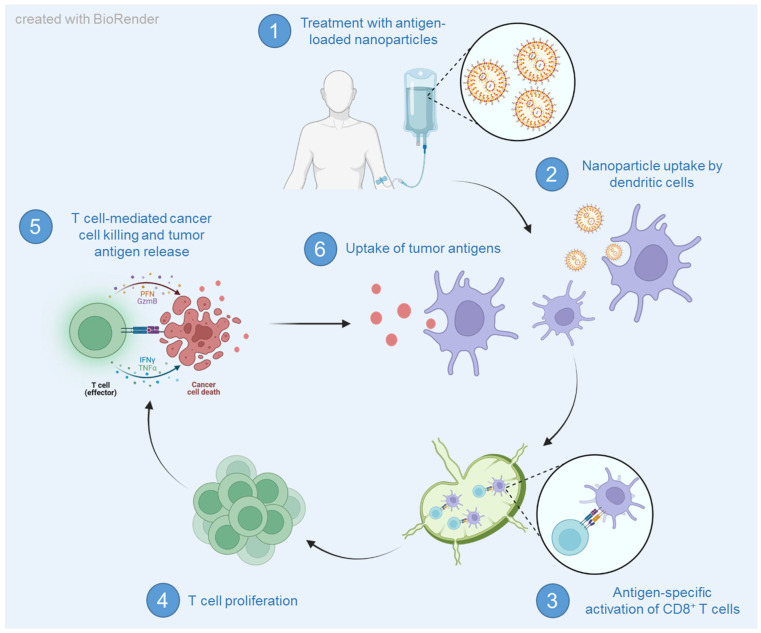
Antigen-loaded nanoparticle vaccines induce specific cancer cell killing. Depicted is the principle of nanoparticulate nanovaccines, which specifically transport antigens in the form of peptides, DNA or RNA to dendritic cells and thereby induce T cell activation.

## Data Availability

Not applicable.

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
