# Peer review of "Delivery of Immunostimulatory Cargos in Nanocarriers Enhances Anti-Tumoral Nanovaccine Efficacy"

_ijms, 2023, doi:10.3390/ijms241512174_

Round 1
Reviewer 1 Report
The manuscript reviews critical knowledge regarding the immunosuppressive nature within tumor microenvironments and different major types of cancer vaccines, showcasing how encapsulation of immunostimulatory cargos in nanoparticles can benefit subsequent delivery of the vaccine into antigen presenting cells that elicit anti-tumor response. This topic is of significant importance and the field benefits from frequent and timely review of the state-of-art progresses. It is certainly suitable to be published in International Journal of Molecular Sciences. The overall logics and structure of this manuscript is satisfying; however, I have reservations about some aspects of the manuscript as summarized below. I believe the manuscript will be outstanding and more suitable for publication after these points are addressed.
Some of the major points are:
** The manuscript lacks discussions over the most recent, and also promising system of mRNA lipid nanoparticles as vaccines. I would suggest add lipid nanoparticles as a subset in Fig. 2 and put some efforts in discussing some recent advances. In such case, immunostimulatory effects can come from the immunogenic nature of unmodified mRNA, or come from the lipid carriers that have immunostimulatory nature (see an example here: 10.1038/s41587-019-0247-3). Some recent news regarding confirmed efficacy of mRNA vaccine in combination with anti-PD-1 may be great in discussing the topics in this manuscript, such as in section 7 – I think this section is now too thin in its content.
** The descriptions on page 5, line 209 to 211 that DC uptake cannot be influenced by NP properties are too arbitrary. See an example here on NP size-dependent DC cell uptake: 10.1016/j.jconrel.2022.06.017.
** The descriptions on page 6, line 232 to 234, read unclear to me. Please elaborate on “nanospheres made of carbon”.
** Related to the descriptions on page 7, line 250 to 252: a very important aspect of the immune-stimulatory effects by single-stranded RNA is its unmodified or modified nature in the uridine. It serves as one of the foundations of mRNA vaccines and should be discussed.
** The discussions on page 8, line 326 to 338, in which chemotherapy drugs are the sole functional payloads of nanoparticles, seem to be less relevant to the topic of this manuscript.
** In the Conclusions, it would be great to give some perspectives on future directions.
Here are some of the minor points that would improve the discussions in this manuscript:
** On page 2, line 64: it would be better to explain what “to mimic viral infections” means as it refers to concepts that may need deep interpretation.
** One page 3, line 93: the description “to be identified in advance” is confusing.
** For the references listed for page 5, line 206: it is recommended to have more recent references for this point as the nanoparticle delivery field has progressed so much in the past decade.
Reviewer 2 Report
This manuscript deals with the role of nanocarriers for the delivery of Immunostimulatory cargo.
The topic is very interesting, but several changes should be addressed before acceptance of the manuscript.
My comments are:
- The methods of the literature review are missing.
- A table with the advantages and limitations of each class of nanoparticles for the delivery of immunostimulatory cargo should be included.
- Polymeric nanovaccines should be analyzed, too. Which are the main block copolymers used?
- The limitations of scaling up these systems should also be addressed.
- The authors should enrich the manuscript with figures and/or tables.
- A Section for future perspectives should be added.
Round 2
Reviewer 1 Report
I appreciate the efforts by the authors to address my comments to the earlier version of the manuscript. The revised manuscript is of good quality and I therefore recommend publication without need of further revisions.
Reviewer 2 Report
The manuscript can be accepted in its current form.